# De Novo Assembly and Characterization of the Transcriptome of an Omnivorous Camel Cricket (*Tachycines meditationis*)

**DOI:** 10.3390/ijms24044005

**Published:** 2023-02-16

**Authors:** Jun-Hui Lu, De-Long Guan, Sheng-Quan Xu, Huateng Huang

**Affiliations:** College of Life Sciences, Shaanxi Normal University, Xi’an 710119, China

**Keywords:** camel cricket, chemosensory genes, codon usage pattern, transcriptome

## Abstract

*Tachycines meditationis* (Orthoptera: Rhaphidophoridae: Tachycines) is a widely distributed insect in eastern Asia. This species is common in urban environments, and its unique omnivorous diet may contribute to its success in various habitats. However, molecular studies on the species are scarce. Here, we obtained the first transcriptome sequence of *T. meditationis* and performed preliminary analyses to test whether the evolution of coding sequences fits the expectations based on the species’ ecology. We retrieved 476,495 effective transcripts and annotated 46,593 coding sequences (CDS). We analysed the codon usage and found that directional mutation pressure was the leading cause of codon usage bias in this species. This genome-wide relaxed codon usage pattern in *T. meditationis* is surprising, given the potentially large population size of this species. Moreover, despite the omnivorous diet, the chemosensory genes of this species do not exhibit codon usage deviating significantly from the genome-level pattern. They also do not seem to experience more gene family expansion than other cave cricket species do. A thorough search for rapidly evolved genes using the dN/dS value showed that genes associated with substance synthesis and metabolic pathways, such as retinol metabolism, aminoacyl-tRNA biosynthesis, and fatty acid metabolism, underwent species-specific positive selection. While some results seem to contradict the species ecology, our transcriptome assembly provides a valuable molecular resource for future studies on camel cricket evolution and molecular genetics for feeding ecology in insects, in general.

## 1. Introduction

*Tachycines meditationis* (Orthoptera: Rhaphidophoridae: Tachycines), also known as a camel cricket or cave cricket, is widely distributed throughout eastern and central China [1]. In urban environments, the success of *T. meditationis* is comparable to that of cockroaches, making it one of the most commonly observed insects [2]. However, molecular biology studies on this species are scarce [3]. The genomic and functional landscapes of *T. meditationis* and the underlying genetic basis for its adaptation to urban environments remain poorly studied.

In this study, we conducted the first transcriptome sequencing of *T. meditationis*. Although transcriptome sequencing provides less coverage than whole-genome sequencing, it has several advantages, including low cost, efficiency, and targeting, specifically for functional coding sequences (CDS) [4]. Transcriptomic analysis can reveal an organism’s expressed gene sequences, annotations, and expression levels [5,6]. For research that is less concerned with other genomic factors (e.g., repetitive elements), the transcriptome provides sufficient data for probing functional components of a given organism’s genome [7].

We analysed codon usage patterns based on de novo transcriptome assembly [8]. Codon bias has been proven to be related to a variety of biological functions, such as translation molecular mechanisms [9] and tRNA content [10]. Analysing genome-wide codon usage patterns is one of the most efficient and powerful ways to quantitatively measure the influence of different evolutionary forces, such as mutation pressure, selection constraints, and genetic drift in a species [11,12]. The observation that the most common codons in highly expressed genes match the most abundant tRNAs makes codon usage bias a classic example of selection pressure operating at the genome level [13]. Owing to the weak selection for translational efficiency [14], the population-genetic theory predicts that the effect of selection would be more apparent in a larger effective population size. Given that this camel cricket species is widely distributed and successfully adapted to the urban environment, we expect its large effective population size to make the selection an efficient and dominant force in shaping the codon usage pattern.

To characterise gene functions, we annotated genes and estimated their dN/dS values to identify the top genes/pathways that may undergo positive selection [8,15,16]. In particular, we focused on chemosensory genes. Chemosensory genes play a very important role in the daily behavior of creatures, such as predation [17], reproduction [18], and avoidance of natural enemies [19]. In insects, chemosensory genes play a crucial role in identifying volatile and nonvolatile components [20,21], such as sugars and alkaloids in plants. Chemosensory genes help organisms perceive their external environment and respond accordingly [22]. The chemosensory supergene family includes odorant-binding proteins (OBP), chemosensory proteins (CSP), odorant receptors (OR), ionotropic receptors (IR), odorant-degrading enzymes (ODE), sensory neuron membrane proteins (SNMP), and gustatory receptors (GR) [23,24,25]. Previous studies have found that changes in the insect chemosensory system lead to changes in the adaptability of their diet [26]. The German cockroach, *Blattella germanica,* is an example of this theory; it has an extremely omnivorous diet and the most extensive chemosensory gene repertoire known for arthropods, with at least 897 IR and GR genes [27]. *T. meditationis* also has an omnivorous diet, which might contribute to its success indoors. Hence, we expect the evolution of this diet to be associated with the expansion of chemosensory gene families.

Interestingly, neither of the predictions based on camel cricket ecology—strong codon usage bias shaped by selection and expanded chemosensory gene families—is supported by our transcriptome analysis. This highlights how little is known about the genetic background and molecular biology of this species. Future investigations into the population genetics and functional genomics of camel crickets will aid in understanding the variety of molecular mechanisms underlying the processes of insects colonising urban environments.

## 2. Results

We obtained two sets of 9.32 Gb transcriptomic sequencing data for *T. meditationis*. A total of 476,495 raw transcripts, 111,197 unigenes, and 46,593 CDSs were identified. The average lengths of the transcripts, unigenes, and CDSs were 572.3 bp, 368.8 bp, and 488.7 bp, respectively.

The percentages of the four nucleotides in all CDSs were similar: Adenine (A; 26.1%), Thymine/Uracil (T/U; 23.9%), Guanine (G; 25.5%), and Cytosine (C; 24.5%). G and C accounted for 50% of the total nucleotide content. Most CDSs had a GC content between 30% and 70% (Figure 1a). Approximately 1000 CDSs had a GC content of less than 30% or more than 70%, and more than half of the CDSs had a GC content between 50% and 70%. Overall, the CDSs were slightly GC-biased in *T. meditationis*. Figure 1b shows the GC content at three codon positions. The GC content in the third position was higher than that in the first and second positions, and the overall GC content resembled the composition at the first codon position (Figure 1b). In addition, the frequencies of sixteen types of dinucleotides were analysed at different coding positions. The ratio between the observed frequency and the theoretical frequency, which is the mathematical product of the corresponding nucleotide content, was calculated (Figure 1c). Four dinucleotides (CU, GA, GC, and UU) showed relatively higher usage in the first and second positions, with AG, CG, UA, and UG having the lowest usage.

### 2.1. Codon Usage Bias in T. meditationis

At the genome level, no codon (Table 1) has an RSCU (relative synonymous codon usage) value above 1.5 or below 0.5, two thresholds for preferred and avoided codons often used in the literature. The 33 codons with RSCU > 1 (Table 1) were slightly UC-biased at the last base: A (4), G (7), U (10), and C (12). It seems that the CDS in *T. meditationis* preferred codons ending in UC.

This preference could be because of a selection or mutation bias. To disentangle these two factors, we calculated the number of preferred codons using GC-conservative codon pairs and gene expression data (i.e., FPKM values). No GC-conservative codon pairs showed significant usage changes correlating with gene expression—all *p* values were greater than 0.1 (Appendix A), suggesting that the weak codon bias we observed is unlikely due to translational selection.

Our conclusion was further supported by analysis of the variations in codon usage bias among genes. The ENC (effective number of codons) plot showed that the ENC values of most genes were between 40 and 60, indicating a relaxed codon bias, and did not deviate significantly from the value expected based on the GC content of the genes (Figure 2a). The deviation ratios, calculated as (ENC_exp_ − ENC_obs_)/ENC_exp_, mostly fell between 0 and 0.1 (Figure 2b). The correlation between ENC value and gene expression was *p* < 0.001, but with an *R*^2^ = 0.005, suggesting that gene expression can explain very little variation in codon usage bias among genes. Correlations between the ENC values and indices for protein properties (e.g., protein length and amino acid profile) were insignificant. The first two principal components (PC1 and PC2) were used for RSCU values; they accounted for 24% and 6% of the total variation, respectively. The only strong correlation we found was between PC1 and GC3 (R^2^ = 0.9615, *p* < 2.2 × 10^−16)^, suggesting a dominant effect of GC content on codon usage bias (Figure 2c,d).

### 2.2. Evolution of Chemosensory Genes in T. meditationis

Based on annotation, we identified 46 putative OBP genes, 22 CSPs, 38 ORs, 13 IRs, 6 SNMPs, and 5 GRs in *T. meditationis* (Figure 3a). There were more genes in OBP and OR than in GR and SNMP, consistent with the sizes of these gene families in most insects [7,26]. The ENC and GC12/GC3 values of the six gene families in *T. meditationis* were further compared with those of the whole gene sequence, and the results showed that the codon bias of the chemosensory genes did not differ from the codon bias of the whole genome (Figure 3b).

Blasting the transcripts against two other cave cricket transcriptomes, we obtained a similar profile in the other two species: 109 putative chemosensory genes in *Ceuthophilus sp.* (37O BPs, 23 CSPs, 32 ORs, 8 IRs, 6 SNMPs, and 3 GRs) and 105 putative genes in *Neonetus sp*. (41O BPs, 21 CSPs, 31 ORs, 4 IRs, 6 SNMPs, and 2 GRs).

We constructed a phylogenetic tree with the OR gene sequences of *T. meditationis*, two species of camel crickets (*Ceuthophilus sp.* and *Neonetus sp.*), and two species of locust (*Ceracris kiangsu* and *Locusta migratoria* [28]; Figure 4). The migratory locust (Table 2) had the largest repertoire, which is unsurprising, as this species has an assembled whole-genome sequence. The results showed that most of the OR genes of *T. meditationis* and camel crickets clustered together, suggesting a contrast between the two suborders. There are several species-specific OR clusters for *T. meditationis*; however, this species does not have more clusters than the two cave crickets (Figure 4). We found similar results for five other gene families (Appendix A).

### 2.3. Genes under Positive Selection in T. meditationis

Including transcriptome datasets from two other species of cave crickets, OrthoFinder [30] identified 1186 single-copy orthologous genes across the three species. After translational alignment, the dN/dS, dN, and dS values for each gene were calculated in PAML using the free-ratio model (M1), which allowed all species branches to have independent values. The distribution of dN/dS of *T. meditationis* is shown in Appendix A—most genes are less than 1. In the *T. meditationis* branch, we identified 124 genes with dN/dS > 1. We conducted KEGG enrichment analysis of these 124 genes and found only seven pathways. The most enriched pathways were the metabolism of xenobiotics by cytochrome P450, other types of O-glycan biosynthesis, retinol metabolism, ascorbate and aldarate metabolism, and drug metabolism-cytochrome P450 (Figure 5a).

The free-ratio model assumes one dN/dS is shared across all sites in a gene. However, positive selection might only affect a few sites along particular lineages. Hence, we employed the branch-site models in PAML to scan for genes that might be affected by positive selection on a small proportion of sites on the *T. meditationis* branch. One hundred twenty-two genes significantly differ between Model A and the corresponding null model (*p* < 0.05). We further performed KEGG enrichment analysis with these genes. We found twenty-three related pathways, mainly concentrated in one carbon pool by folate, biosynthesis of unsaturated fatty acids, mitophagy-animal, pantothenate, and CoA biosynthesis (Figure 5b).

## 3. Discussion

*T. meditationis* is a widely distributed species of camel crickets in southeastern China. These nocturnal insects can reside outdoors in dark and damp environments, are often observed indoors, and feed on food debris. Hence, it earned the Chinese common name “Zhao Ma” (stove horse). Despite its abundance in urban environments, studies on this species are limited. Most studies have focused on the taxonomy of this group. In this study, we assembled and annotated a transcriptome dataset of *T. meditationis*, which is the first genome-scale dataset for this species. Although the transcriptome only provides information on expressed genes at the time of the experiment, this rich dataset can be used for various analyses. In this study, analyses of codon usage and chemosensory gene families revealed unexpected results. Below, we discuss these results, possible caveats, and future research directions.

### 3.1. Weak Codon Usage in T. meditationis

Codon usage—the frequency of synonymous codons—has been the subject of molecular evolution for decades [31,32]. The match between codon frequencies and tRNA abundance in the cell has been well documented in microbial genomes [33] (e.g., *Escherichia coli* and *Saccharomyces cerevisiae*) since the 1980s. Advances in molecular engineering techniques have further enabled researchers to demonstrate that codon choices can affect the translational efficiency and accuracy of genes [34]. In addition to selection pressure from the translation process, mutation pressure, mutation biases or GC-biased gene conversion favouring G and C over A and T alleles, can also contribute to codon usage bias [35]. Transcriptome data provide the exact information needed to disentangle the evolutionary forces behind codon usage bias—the CDS for calculating codon frequencies and gene expression levels to assess the influence of translational selection.

We found that codon usage bias is weak in *T. meditationis,* in general. Most of the variation among the genes can be explained by the GC content at the third codon position, which is most likely due to mutation pressure. Eukaryotic genomes lacking a strong intensity of codon usage bias are common. Some have codon usage that is only weakly, or not at all, correlated with the expression level of genes or tRNA abundance (e.g., in humans [36]). Several hypotheses have been proposed to explain this variation among species [37]. Insight from population genetics has shown translational selection to be so weak that it becomes effective only in species with a large effective population size [38]. In species with a small effective population size, nonoptimal codon choices are only mildly deleterious and can still be fixed by genetic drift. Analyses across species reported higher bias in eukaryotes with shorter generation times, presumably species with large *Ne* (effective population size) [39] and more codons that have usage correlating with gene expression in large *Ne* animals [40].

Surprisingly, we found a very weak codon usage bias (Table 1) in *T. meditationis*,a commonly observed insect species with a short generation time and wide distribution range. In fact, excluding mutation pressure in examining only the conserved GC codon pairs, we found no preferred codon. One possible explanation is the decoupling of the effectiveness of this species and census population size. The population size (and range) of this species might have only recently expanded as humans provided a new habitat type; the low effective population size reflects the genetic diversity before the expansion. Another possibility is that *T. meditationis* harbours a deep population structure, which can also reduce the effective population size. According to the Orthoptera Species File, the taxonomy of the genus *Tachycines* has quickly evolved in the past few years—more than 30 new species have been reported since 2018 according to the Orthoptera Species File [41]. The distribution range of *T. meditationis*, its population dynamics, and possible cryptic species diversity would be worth examining with a phylogeographic approach [42].

### 3.2. Chemosensory Gene Evolution in Camel Crickets

Similar to other animals, chemoreception is critical for insect survival. Chemosensory genes are crucial for foraging, sensing food sources, and preventing poisoning. They also play a role in inter- and intra-species communication, such as mate choice and predator avoidance. The evolution of chemosensory genes is an excellent system to investigate how species adapt to new environments. Studies have found that chemosensory genes can respond rapidly to selection. Among the global lines of *Drosophila melanogaster*, chemosensory gene families show the strongest selection signals in genome-wide analyses [22]. Many studies have shown that chemosensory genes have played a crucial role in adapting to the environment [43,44]. For example, studies revealed a large-scale expansion of the chemosensory genes in German cockroaches [45], suggesting a vital role for chemosensory gene evolution in adapting to a wide range of climates and food sources during the global expansion of this species [44]. Comparisons across sister species of orchid bees also detected elevated signals of divergent selection among chemosensory receptors [46]. In addition to sequence evolution, gene family sizes of chemosensory genes have been correlated with diet breadth. For example, host-specific beetle species have fewer chemosensory genes than polyphagous species do [47]. Comparisons of different species of mosquitos and *Drosophila* discovered a similar trend [48,49].

Given the omnivorous diet and presumably recent adaptation to the urban environment, we expected *T. meditationis* to have enlarged species-specific chemosensory gene repertoires, especially for OR and IR genes. However, all three camel cricket species in the family Rhaphidophoridae had similar repertoire sizes. While some species-specific gene clusters were present in the gene family tree, *T. meditationis* did not seem to have more gene duplication events than the other two species (Figure 4a). It should be noted that transcriptome data are limited by the specific developmental stage and tissue used for mRNA extraction. For example, the bamboo locust (*Ceracris kiangsu*) transcriptome data from the antenna has a much higher proportion of ORs (91 ORs, 13 IRs, 13 OBPs, 6 CSPs, and 2 SNMPs) [50]; hence, some OR genes might be missing from the transcriptome data (Table 2). A thorough comparison of gene family sizes among species requires whole-genome sequencing.

Here, we focused on the one feature of *T. meditationis*—its omnivorous diet. However, this species also has other relatively uncommon characteristics in the suborder Ensifera, such as apterous and lacking acoustic communications. Comparing its genomic data with other cricket species’ genomes [51,52] could provide insights into how these creatures sense the environment and communicate among individuals.

## 4. Materials and Methods

### 4.1. T. meditationis Transcriptome Sequencing and Assembly

The two *T. meditationis* specimens used in this study were collected in Xian, Shaanxi, China and stored in liquid nitrogen until mRNA extraction. Using the NEBNext UltraTM DNA Library Prep Kit (New England Biolabs, Ipswich, MA, USA), two cDNA libraries (T1 and T2) were constructed from two *T. meditationis* individuals following the manufacturer’s instructions and sequenced on an Illumina HiSeq 2500 high-throughput sequencing platform (Illumina, San Diego, CA, USA). Raw sequencing data were uploaded to the NCBI database (project number: PRJNA912169, SRA number: SRR22734937).

After quality filtering, reads from each library were assembled in Trinity 2.4.0 (http://trinityrnaseq.github.io/ (accessed on 21 July 2020) with default parameters [53]. The transdecoder function was used to extract transcripts with open reading frames (ORF) [54]. Using BLAST (https://blast.ncbi.nlm.nih.gov/Blast.cgi (accessed on 28 July 2021), we compared the transcripts assembled from the two libraries and only retained those that exist in both libraries (i.e., having an blastx match in the other library with an E-value < 1 × 10^−50^) [55].

### 4.2. CDS Identification and Annotation

Unigenes were annotated by blasting against the following five databases: NR (NCBI nonredundant protein sequences) [56], Pfam (Protein family) [57], COG/egg-NOG (clusters of orthologous groups of proteins) [58], KEGG (Kyoto Encyclopedia of Genes and Genomes) [59], and GO (Gene Ontology) [60]. The BLAST parameters x, p, and r were set with an E-value threshold of 1 × 10^−5^. Unigenes with a BLAST match in at least one database were used for further analyses. We ran ORFfinder (http://www.Geneinfinity.org/sms/sms (accessed on 5 August 2020) on these sequences to confirm the existence of the CDS. To ensure the quality of annotated genes, we deleted sequences shorter than 200 bp, containing more than 10% internal N gaps, or having more than one internal stop codons.

### 4.3. Codon Usage Analysis

Nucleotide composition of the whole transcriptome was determined using DAMBE http://dambe.bio.uottawa.ca/DAMBE/dambe.aspx (accessed on 10 August 2020) [61]. Compositions at each codon position and dinucleotide composition were calculated using CodonW. The correlation between GC12 and G3 was examined [62].

We calculated the RSCU value and ENC using the standard codon table in CodonW to quantify the codon usage bias for each CDS [63]. RSCU values show the exact frequency of each synonymous codon in the CDS compared to the expected frequency if all codons for an amino acid are used equally. Hence, they provide the most detailed information on codon usage for each CDS. The ENC reflects the extent to which codon usage deviates from random selection [64]. That is, each CDS has one ENC value and 61 RSCU values (64 codons, excluding the 3 stop codons).

### 4.4. Genome-Wide Codon Usage Bias

We identified which codons were overrepresented or underrepresented at the genome level by calculating the summary RSCU value of each codon. To examine the similarity between different codons, we conducted principal component analysis (PCA) on the codons’ RSCU matrix, in which rows and columns correspond to codons and CDSs, respectively.

Next, we estimated the number of preferred codons defined in terms of gene expression—those used more frequently in genes with high expression than in genes with low expression [65]. We analysed GC-conservative pairs of synonymous codons to exclude the potential influence of GC-biased conversion [66], that is codon pairs that are XYA/XYT or XYC/XYG; there are 17 such pairs in the standard genetic code. GC-biased conversion did not affect the relative frequency of the two codons in a pair. We calculated the relative usage of 17 codon pairs for each gene and correlated it with gene expression. Following Galtier et al. (2017) [40], we defined preferred codons as those with an absolute correlation coefficient (r) > 0.05 and *p*-value < 0.001.

### 4.5. Variation of Codon Bias among Genes

We explored the variation in codon usage bias among CDS to investigate the evolutionary forces responsible for codon bias in *T. meditationis.* When mutation pressure is the only force acting on the third codon position, GC3 should determine the gene’s level of codon bias [67]. In contrast, codon bias should correlate with the expression level with translational selection, as highly expressed genes should be under stronger selection for translation efficiency [63].

Hence, for ENC, we generated an ENC plot, which is a scatter plot showing the association between the ENC value and GC3 (G or C content in the third codon position). The expected ENC (expENC) based on GC3 was obtained by submitting 1000 GC values equally spaced between 0.001 and 1.000 to DAMBE [68]. The smooth curve connecting these expected values was compared to the observed ENC from the empirical CDS. We then tested the correlation between ENC and gene product properties. In addition to the gene expression level, we obtained the amino acid composition, aromaticity (AROMO) value, and hydrophobicity (GRAVY) value using CodonW (https://sourceforge.net/projects/codonw (accessed on 12 August 2020) for each CDS. The GRAVY and AROMO values reflect the percentage of hydrophobic and aromatic amino acids in proteins that affect protein folding. We used principal components (PCs) to represent high-dimensional amino acid composition data.

Similar analyses were conducted for the RSCU values. We performed PCA to reduce the dimension of the data (each gene had 61 RSCU values) and correlated the first two PCs with gene properties (e.g., GC3, expression level, and GRAVY).

### 4.6. Chemosensory Gene Analysis

Based on annotation information, we identified genes belonging to six chemosensory gene families. We examined the codon bias of these genes. To study gene family evolution, we downloaded the chemosensory genes identified in bamboo grasshoppers (*Ceracris kiangsu;* accession number: SRR1648029). We also searched NCBI for additional transcriptome datasets for the Rhaphidophoridae (cave crickets) family and identified two datasets: *Neonetus sp.* AD-2015 (cave weta; accession number: SRR2230582) and *Ceuthophilus sp.* AD-2013 (another camel cricket species; accession number: SRR921579). Unigenes were assembled and identified using the same approach as that used for our dataset. Chemosensory genes in these two cave cricket datasets were identified by blasting the chemosensory genes in our dataset (E-value < 1× 10^-10^).

Sequences of each gene family were pooled, and sequence alignments were obtained for each gene family using Mafft [69] with the default setting. Phylogenetic trees were constructed using IQtree [70] based on the GTR + F model and visualised in FigTree (v1.4.3) (http://tree.bio.ed.ac.uk/software/figtree/ (accessed on 1 August 2020)). The phylogenetic tree was rooted using the default IQtree parameters.

### 4.7. dN/dS Analysis to Identify Genes under Positive Selection

Orthologous gene sequences from closely related species can help identify genes that have experienced species-specific positive selection during evolution. Orthofinder [30] with default parameters was used to identify orthologous genes between *T. meditationis* and the other two cricket species. Alignments of single-copy orthologous genes from three species were generated by Mafft [69], and bmge (g: 0.5, the value of g indicates the maximum proportion of gap and columns; genes exceeding this proportion are deleted) were used to prune the result of multiple sequence alignment [71].

The *codeml* program in PAML [72] was used to test for positive selection. First, we fitted the free-ratio model (M1) to calculate the branch-specific ω values for each species. Genes with ω > 1 for *T. meditationis* were selected for further gene set enrichment analysis using KOBAS [73]. Since positive selection might only affect a few sites along particular lineages, we also applied a second test, using branch-site models that allow ω to vary between sites in the gene and across branches on the phylogeny [74]. Specifically, we compared two ML models in the second test: (1) a null model assuming two site classes, one with dN/dS  <  1 and one with dN/dS  =  1 (model  =  2, NSsites  =  2, fix_omega  =  1, omega  =  1), and (2) an alternative model with one site class having dN/dS  >  1 (model  =  2, NSsites  =  2, fix_omega  =  0). The branch leading to *T. meditationis* was selected as the foreground, whereas the others were kept as the background. The fit of the two models was compared using a likelihood ratio test with one degree of freedom, and a gene set enrichment analysis was conducted with significant genes.

## 5. Conclusions

This study characterised a de novo transcriptome of *T. meditationis*, an insect commonly observed in urban environments. Contrary to expectations grounded in the ecology and diet of this species, we did not find evidence for intense selection of codon usage or enlargement of the chemosensory gene repertoires. However, in combination with the transcriptomes of other species of the same family, we identified genes that might undergo species-specific positive selection and enrich biological pathways. Our dataset is a valuable molecular resource for further improving our understanding of the evolution of camel crickets, and our results will facilitate future studies on how insects’ diet adapts to urban environments in general.

## Figures and Tables

**Figure 1 ijms-24-04005-f001:**
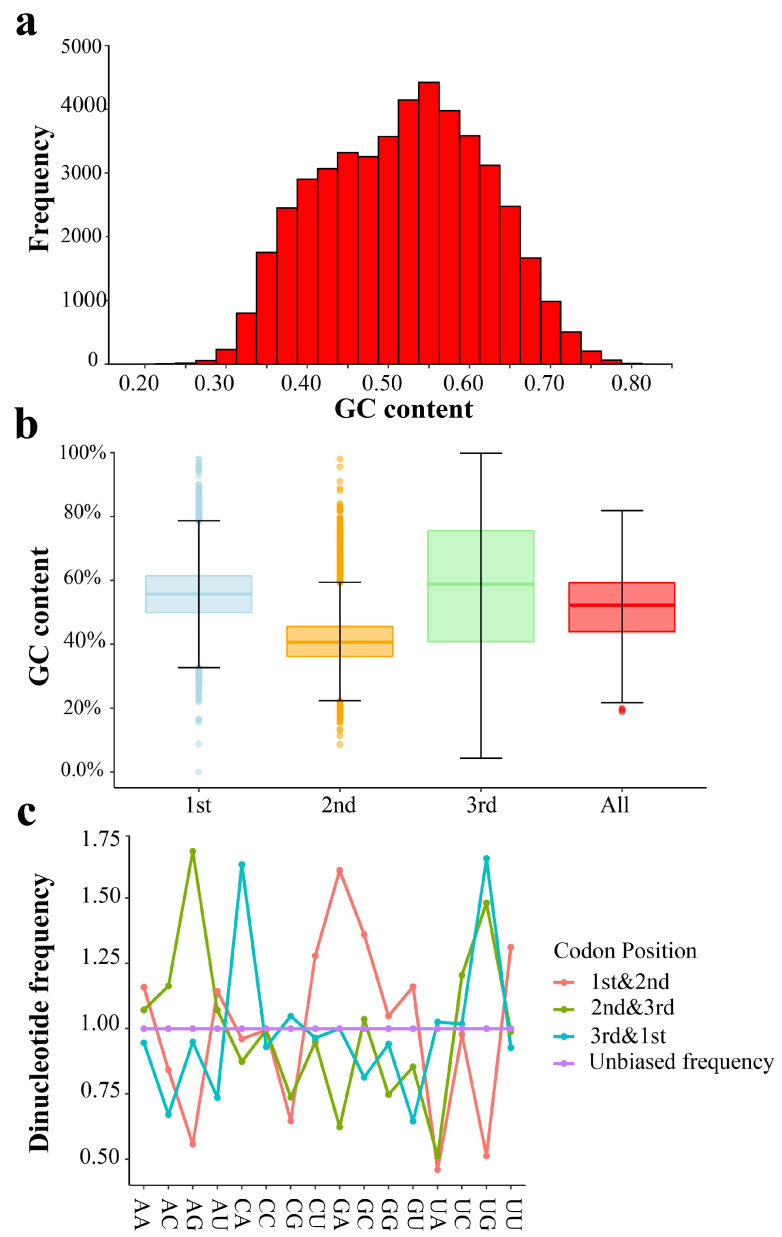
Base composition of the assembled transcriptomic data. (**a**) Distribution of GC content levels among coding sequences (CDS). (**b**) Box plot showing GC content variation among different codon positions and the genome average (All). (**c**) Dinucleotide frequencies at different codon position combinations (1st and 2nd, 2nd and 3rd and 3rd and 1st). The straight line represents the theoretical equal frequency (1/16).

**Figure 2 ijms-24-04005-f002:**
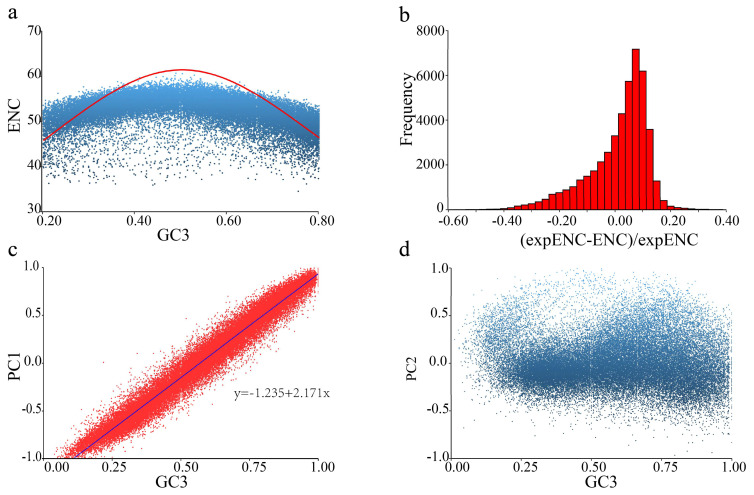
Codon usage bias of CDS. (**a**) The ENC-GC3 plot. GC3 is the GC content at the third codon position, and the solid line represents the expected curve when codon usage bias is only affected by mutation pressure. (**b**) Distribution of deviations from the expected ENC. (**c**) Correlation of PC1 and GC3. (**d**) Correlation of PC2 and GC3. PC1 and PC2 are the first two principal components of the codons’ RSCU matrix.

**Figure 3 ijms-24-04005-f003:**
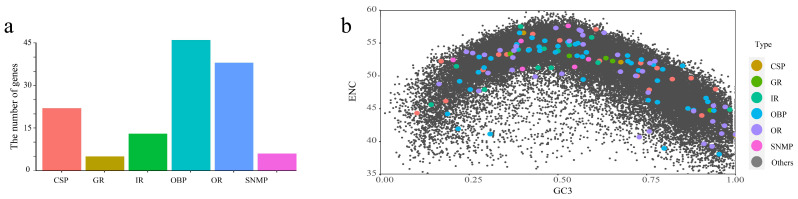
Codon usage bias of chemosensory genes in *T. meditationis*. (**a**) Number of identified chemosensory genes. Odorant-binding protein (OBP), chemosensory protein (CSP), odorant receptor (OR), ionotropic receptor (IR), sensory neuron membrane protein (SNMP), and gustatory receptor (GR). (**b**) The ENC-GC3 plot of chemosensory genes. Chemosensory genes are coloured according to (**a**). The gray points represent other CDS.

**Figure 4 ijms-24-04005-f004:**
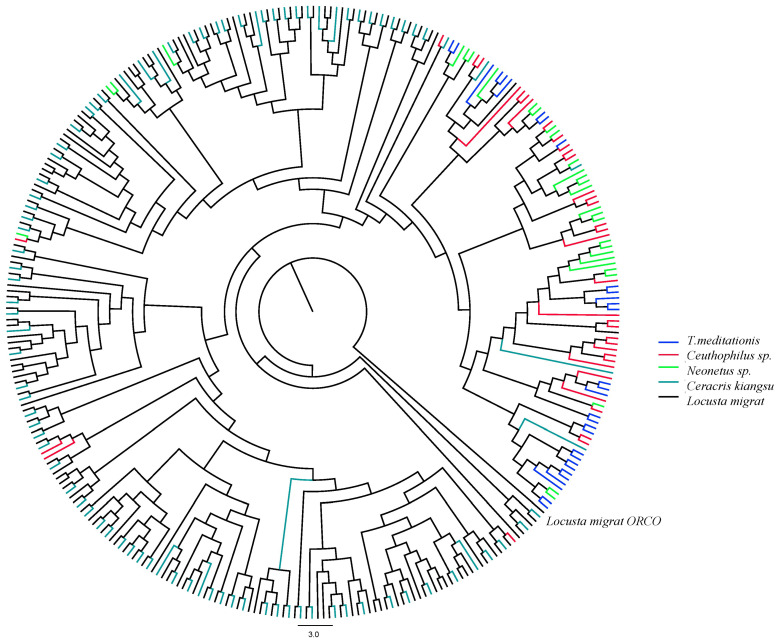
A maximum-likelihood phylogenetic tree of the OR gene family. Different colours represent different species of Orthoptera. The tree is rooted with the ORCO gene in *Locusta migratoria*.

**Figure 5 ijms-24-04005-f005:**
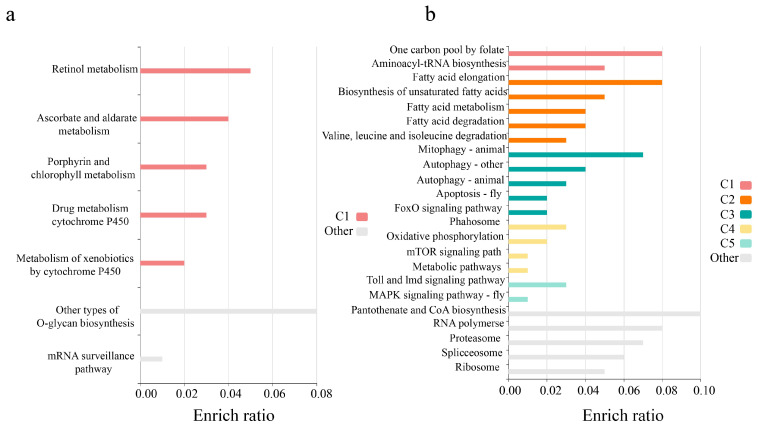
Results of KEGG enrichment analysis genes undergone species-specific positive selection in *T. meditationis*. Genes were identified by (**a**) the free-ratio model or (**b**) the branch-site model in PAML.

**Table 1 ijms-24-04005-t001:** Overall relative synonymous codon usage (RSCU) values of all CDSs in *T. meditationis*.

AA	Codon	RSCU	AA	Codon	RSCU	AA	Codon	RSCU	AA	Codon	RSCU
Ala	GCU	1.122	His	CAC	1.003	Pro	CCA	1.076	Ser	UCG	0.768
	GCG	0.687		CAU	0.997		CCC	1.014		UCU	1.187
	GCC	1.301	Ile	AUU	1.190		CCU	1.182	Thr	ACC	1.223
	GCA	0.891		AUA	0.465		CCG	0.727		ACA	1.021
Cys	UGU	1.010		AUC	1.345	Gln	CAA	0.849		ACG	0.674
	UGC	0.990	Lys	AAA	0.784		CAG	1.151		ACU	1.082
Asp	GAU	1.023		AAG	1.216	Arg	AGA	1.187		GUU	1.048
	GAC	0.977	Leu	CUA	0.392		AGG	0.813		**GUG**	**1.282**
Glu	GAG	1.067		CUC	1.148		CGA	0.772		GUC	1.126
	GAA	0.933		**CUG**	**1.478**		CGC	1.460		GUA	0.545
Phe	UUU	0.847		CUU	0.982		CGG	0.668	Trp	UGG	1.000
	UUC	1.153		UUA	0.793		CGU	1.100	Tyr	UAC	1.083
Gly	GGU	1.153		UUG	1.207	Ser	AGC	1.060		UAU	0.917
	GGG	0.519	Met	AUG	1.000		AGU	0.940			
	**GGC**	**1.388**	Asn	AAC	1.062		UCA	1.048			
	GGA	0.941		AAU	0.938		UCC	0.998			

Notes: All codons except for stop codons were included in this analysis. AA: amino acids. Codons with RSCU > 1.25 are shown in bold, codons with RSCU < 0.75 are underlined.

**Table 2 ijms-24-04005-t002:** The number of chemosensory genes of four orthoptera insects.

Species	OR	GR	IR	CSP	OBP	SNMP
*Oedaleus asiaticus* [29]	60	-	6	-	15	3
*Ceracris nigricornis*	71	-	8	10	20	3
*Tachycines meditationis*	38	5	13	22	46	6
*Locusts migratoria* [28]	142	28	32	-	-	-

## Data Availability

RNA-Seq data can be found with accession number PRJNA912169. The RNA-Seq data is publicly available on National Center for Biotechnology Information.

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
