# Peer review of "De Novo Assembly and Characterization of the Transcriptome of an Omnivorous Camel Cricket (Tachycines meditationis)"

_ijms, 2023, doi:10.3390/ijms24044005_

Round 1
Reviewer 1 Report
This group of insects occupies characteristic niche but uderstudied. Since the sugenera are unique to each continent, phylogenetic origin is old. No species is known for producing a flying form, i.e., they are apterous, hiding during daytime. Thease features must be characterized by comparing genome databases of spcies. The systems that effectively depict these characteristics include 1)camel drickets belonging to other subgenus, 2) myrmecophilids and 3)alary crickets. Since the species lacks all acaustic communication tools. it is right to examine odorant receptors and alias.
Kataoka,K.. R.Minei , K. Ide, A. Ogura, H. Takeyama, M. Takeda, T. Suzuki , K.Yura and T.Asahi (2020) The draft genome dataset of the Asian cricket Teleogryllus occipitalis for molecular research toward entomophagy. Frontiers in Genetics, 11 | Article 470
Sanno,R., Kataoka K, Hayakawa S, , Keigo Ide K., Nguyen CN, Nguyen TP, . Le BTN, Kim QTP , Mineta K , Takeyama H, Takeda M, T SatoT, Suzuki T,, Yura K, , Toru Asahi1 Comparative analysis of mitochondrial genomes in Gryllidea (Insecta: Orthoptera): Implications for adaptive evolution inant-loving crickets. Genome Biology and Evolution. 13(10) doi:10.1093/gbe/evab222
Author Response
Dear Reviewer:
Thank you for taking the time to review our paper. According to your advice, we used a professional English editing service to improve the English language of our manuscript.
Regarding the genome databases of related species, our analysis used two species: Ceuthophilus sp. and Neonetus sp., from the same superfamily Rhaphidophoroidea. These two species are the only ones in this superfamily with transcriptomic datasets deposited in the NCBI SRA databases. There were no transcriptome data available for Myrmecophilidae. The Asian cricket Teleogryllus occipitalis belongs to another superfamily Grylloidea, which is very distantly related to our species (more than 260Mya of divergence; Song et al., 2015), so we did not include it in our analysis. Yet, we agree with the reviewer that comparing genomes of apterous cricket could help us discover shared mechanisms of how these species lacking acoustic organs sense the environment and communicate among individuals. We added these to our discussion and cited the two articles (Line279).
Please let us know if you have any questions regarding this submission or our response to reviewer comments. We look forward to hearing from you.
Best regards,
Jun-hui Lu, De-long Guan, Sheng-quan Xu and Huateng Huang

Reviewer 2 Report
To,
The Editor,
IJMS, MDPI,
Manuscript ID: ijms-2170599
Subject: Submission of comments on the manuscript in “IJMS"
Dear Editor IJMS, MDPI,
Thank you very much for the invitation to consider a potential reviewer for the manuscript (ID: ijms-2170599). My comments responses are furnished below as per each reviewer’s comments.
Dear Chief Editor,
The present work authors transcriptome sequencing of T. meditationis, retrieving 476,495 effective transcripts and annotating 46,593 coding sequences (CDS). We analyzed the codon usage and found that rather than selection, directional mutation pressure was the leading cause of codon usage bias in this species. This genome-wide relaxed codon usage pattern in T. meditationis is surprising given this species' potentially large population size. Despite the omnivorous diet, this species' chemosensory genes do not have a codon usage deviating significantly from the genome-level pattern. They also do not seem to experience more gene family expansion than other cave cricket species. A thorough search for rapidly evolved genes using the dN/dS value showed that genes associated with substance synthesis and metabolic pathways had undergone species-specific positive selection, such as retinol metabolism, aminoacyl-tRNA biosynthesis and fatty acid metabolism. While some of the analysis results seem to contradict the species' ecology, our transcriptome assembly provides a valuable molecular resource for future study on camel crickets' evolution and molecular genetics for feeding ecology in insects in general. The manuscript represents a very important piece of research in a logical presentation. Therefore, it might be conditionally accepted as subject to major revision. Instead, authors have to improve their manuscripts with many non-clear meanings, inaccuracies, and the authors need to address the following issues before it can be accepted for publication.
- I have read the entire manuscript and my initial comment is that manuscript is poorly written. I have significant concerns about the grammar and vocabulary of the manuscript; therefore, I recommend the authors to used an English proofreading service..
- The structure of the abstract should be improved, as well as the lack of several aspects that should be included in this section. Most of the abstracts contain confusing and uninformative sentences. Please give more precise objectives here (such as in the Abstract). The abstract should highlight the most important results of the parameters and characteristics assayed.
- Keyword must in alphabtical order.
- Introduction grammatical issues appear to be most prevalent in the introduction, making for very confusing reading. Further, the introduction is short but has no clear thread.
- Why you selected this crop for your experiment? Please provide the detail of the used variety.
- The figures are quite low resolution and difficult to make out. Higher-resolution versions will be needed for publication. Further, text in figure is not readble. for example, in Figures 1, 2, 3, 4, and 5.
- In Material and Methods:- indicate how many replicates assayed in each analysis/parameter. The number of samples or biological and technical replicates should be mentioned for each parameter in the methods.
- The discussion should be interpreted with the results as well as discussed in relation to the present literature.
Author Response
Dear Reviewer:
Thanks for your letter and comments concerning our manuscript. We have carefully considered and made responses to all the comments.
- I have read the entire manuscript and my initial comment is that manuscript is poorly written. I have significant concerns about the grammar and vocabulary of the manuscript; therefore, I recommend the authors to use an English proofreading service.
> According to your suggestions, we have chosen a professional English proofreading service to modify the grammar and vocabulary of our article. We uploaded the certificate that the service provided in the system.
- The structure of the abstract should be improved, as well as the lack of several aspects that should be included in this section. Most of the abstracts contain confusing and uninformative sentences. Please give more precise objectives here (such as in the Abstract). The abstract should highlight the most important results of the parameters and characteristics assayed.
> Thank you for your suggestions. We have added the objectives in the Abstract (Line 8-Line 9).
- Keyword must in alphabetical order.
> We have sorted the keywords in our paper in alphabetical order.
- Introduction grammatical issues appear to be most prevalent in the introduction, making for very confusing reading. Further, the introduction is short but has no clear thread.
> Thank you for raising this issue. As mentioned above, we used a professional language editing service to proofread our manuscript for grammatical problems. The introduction has five paragraphs, and this is how they are organized: the first paragraph introduces the studied species; the second one is about why we chose transcriptome sequencing; the third and fourth paragraphs present the analyses we performed with transcriptomic sequence data— analyses of codon usage bias and sequence evolution—and the expected results (i.e., null hypotheses) given what we know about the species; and the introduction ends with the last paragraph giving a brief summary of the study results.
- Why you selected this crop for your experiment? Please provide the detail of the used variety.
> We do not fully understand this question. Tachycines meditationis (Orthoptera: Rhaphidophoridae: Tachycines) is an insect commonly known as camel cricket. We guess that the question is about why we choose this species? This species is widely distributed throughout eastern and central China and commonly observed in the urban environment, but molecular studies on species are scarce, but are needed for further understanding this species’ ecology and evolutionary history. As far as we know, there is no subspecies named for this species. We provided information about where the specimens were collected in the Materials and Methods.
- The figures are quite low resolution and difficult to make out. Higher-resolution versions will be needed for publication. Further, text in figure is not readble. for example, in Figures 1, 2, 3, 4, and 5.
> We apologize for the low resolution and small text in the figures. We modified the size of the text in the picture and made sure that the resolutions of all the figures are 600dpi.
- In Material and Methods: indicate how many replicates assayed in each analysis/parameter. The number of samples or biological and technical replicates should be mentioned for each parameter in the methods.
> Thanks for pointing out this mistake. We sequenced two specimens (info added to Line294).
- The discussion should be interpreted with the results as well as discussed in relation to the present literature.
> Thank you for y raising this issue. We have now added more sentences discussing the results in the context of recently published literature (Line215, Line253-255, Line276-280).
Please let us know if you have any questions regarding this submission or our response to reviewer comments. We look forward to hearing from you.
Best regards,
Jun-hui Lu, De-long Guan, Sheng-quan Xu and Huateng Huang

Round 2
Reviewer 1 Report
The authors made satisfactory revision and the manuscript is ready to be published as this is.
Reviewer 2 Report
Dear Editor,
Thank you for providing the opportunity to review the revised manuscript. The manuscript is improved considerably after revision according to the reviewer's comment. Now this study is a suitable contribution to the IJMS. I recommend the manuscript for publication.
Thank you
With best regards